# MoTE: Mixture of Ternary Experts for Memory-efficient Large Multimodal Models

## Abstract

Large multimodal Mixture-of-Experts (MoEs) effectively scale the model size to boost performance while maintaining fixed active parameters. However, previous works primarily utilized full-precision experts during sparse up-cycling. Despite they show superior performance on end tasks, the large amount of experts introduces higher memory footprint, which poses significant challenges for the deployment on edge devices. In this work, we propose **MoTE**, a scalable and memory-efficient approach to train **M**ixture-**o**f-**T**ernary-**E**xperts models from dense checkpoint. Instead of training fewer high-precision experts, we propose to train more low-precision experts during up-cycling. Specifically, we use the pre-trained FFN as a shared expert and train ternary routed experts with parameters in {-1, 0, 1}. Extensive experiments show that our approach has promising scaling trend along model size. MoTE achieves comparable performance to full-precision baseline MoE-LLaVA while offering lower memory footprint. Furthermore, our approach is compatible with post-training quantization methods and the advantage further amplifies when memory-constraint goes lower. Given the same amount of expert memory footprint of 3.4GB and combined with post-training quantization, MoTE outperforms MoE-LLaVA by a gain of 4.3% average accuracy on end tasks, demonstrating its effectiveness and potential for memory-constrained devices.[1]

## 1 Introduction

Large Multimodal Models (LMMs) (Abdin et al., 2024; McKinzie et al., 2024; Zhang et al., 2024a; Wang et al., 2024d; Chen et al., 2024b; Bai et al., 2025) have achieved remarkable performance across a wide range of downstream tasks, including visual question answering and autonomous computer agents. However, as model size increases, the rising inference cost presents significant challenges for deploying LMMs efficiently. To address this, Mixture-of-Experts (MoE) (Lepikhin et al., 2021; Fedus et al., 2022; DeepSeek-AI et al., 2024) introduces a mechanism that maintains a large pool of experts while activating only a subset for each input, thereby improving computational efficiency. Although MoE models significantly reduce FLOPs, they generally have a higher memory footprint, making deployment on edge devices challenging. For example, when training multimodal MoE up-cycled from Qwen2.5-3B, **if all feed-forward network (FFN) layers are replaced with MoE layers containing 16 experts, the resulting model's non-embedding memory footprint will increase from 5.2GB to 73.2GB.** This limitation is particularly pronounced for consumer-grade GPUs, which often have constrained memory capacities.

Model quantization is a promising approach to reducing the memory footprint of LMMs while maintaining comparable performance. Most mainstream quantization methods (Frantar et al., 2022; Lin et al., 2024b; Chee et al., 2024; Tseng et al., 2024b) aim to compress the bit-width of a pre-trained, full-precision model. Although these methods have a low training cost, they suffer from significant performance degradation when the bit-width is reduced below 4-bit. Recent studies (Ma et al., 2024; Kaushal et al., 2024; Zhu et al., 2024) have demonstrated promising scaling trends for ternary pre-training in Large Language Models (LLMs). At sufficiently large model sizes, ternary models can achieve accuracy comparable to full-precision models on downstream tasks while maintaining the same pre-training cost. Furthermore, they have much lower inference costs in terms of memory, latency, and energy consumption (Wang et al., 2024b). However, since these models have only been

---

[1]We will release the code and model weights for reproducibility.

trained on billions of tokens, a substantial performance gap remains between open-sourced ternary models and full-precision dense models. As a result, directly training MoE models initialized from these under-trained models leads to weak performance on end tasks.

In this work, we introduce **MoTE**, a scalable and memory-efficient architecture designed to train **M**ixture-**o**f-**T**ernary **E**xperts model from a pre-trained, full-precision dense checkpoint in multimodal tuning. Our approach addresses the inefficiency of multimodal MoE models in terms of memory footprint. Prior works (Lin et al., 2024a; Li et al., 2025) primarily replace the FFN layer in dense checkpoints with an MoE layer, initializing the experts using the pre-trained FFN. However, we observed that in ternary training, replacing the FFN layer leads to significant performance degradation, as weight ternarization disrupts the pre-trained FFN. To mitigate this, we retain the FFN from the dense checkpoint as a shared expert activated for all inputs. During up-cycling, the layers inherited from the dense model remain frozen, while only the ternary MoE layers are trainable.

We first conduct strict and controlled experiments to evaluate the proposed approach against full-precision up-cycling MoE-LLaVA (Lin et al., 2024a) across various model scales on a wide range of image understanding tasks. Our results show that ternary up-cycling exhibits surprising effectiveness as model size scales. As the size of the up-cycled dense checkpoint increases, the performance gap between MoTE and MoE-LLaVA narrows, eventually reaching comparable performance at scales larger than 1.5 billion parameters. Additionally, MoTE is compatible with post-training quantization techniques (Frantar et al., 2022). Given the same expert memory footprint and combined with post-training quantization, MoTE outperforms full-precision MoE-LLaVA at both 1.5B and 3B model sizes. This advantage becomes even more pronounced as memory constraints tighten. Specifically, under an expert memory budget of 3.4GB, our approach achieves a 4.3% improvement in average accuracy on downstream task. These results demonstrate that given the same amount of total memory footprint and active parameter counts, training with a larger number of low-precision experts yields better performance than using fewer high-precision experts.

## 2 RELATED WORK

**Mixture of Experts.** LMMs demonstrate superior performance across various tasks as model size and training data scale increase. MoE models (Lepikhin et al., 2021; Fedus et al., 2022; Muennighoff et al., 2024) maintain a large pool of experts but activate only a subset for each token, enabling improved performance at the same FLOPs budget. Komatsuzaki et al. (2023) introduced sparse up-cycling to reduce the training costs of MoE models by initializing them from dense checkpoints. Lin et al. (2024a) explored the up-cycling of LMMs in the context of multimodal training, while Shu et al. (2024) proposed a progressive knowledge transfer strategy to train small-scale multimodal MoEs from dense models. A straightforward way to improve the memory efficiency of MoE models is to train smaller experts or LoRAs (Luo et al., 2024; Wang et al., 2024a). However, since the expert size typically differs from that of the pre-trained FFN, dense checkpoints cannot be directly reused, leading to degraded performance compared with sparse up-cycled MoEs. While prior studies have mainly focused on reducing parameter counts during up-cycling, our work explores an alternative direction, i.e., up-cycling with reduced bit-width.

**Model Quantization.** Quantization is a promising approach to reducing the memory footprint of LMMs while maintaining competitive performance, which can be categorized into two types based on the stage at which it is applied: post-training (Dettmers et al., 2022; Frantar et al., 2022; Lin et al., 2024b; Tseng et al., 2024b) and pre-training quantization (Wang et al., 2025; Peng et al., 2023). Post-training quantization compresses high-precision pre-trained models after training. Due to its lower cost, it is widely adopted for mainstream large-scale models. GPTQ (Frantar et al., 2022) and AWQ (Lin et al., 2024b) reduce the bit-width to 4 bits while incurring minimal degradation. QuIP# (Tseng et al., 2024a) builds on QuIP (Chee et al., 2024) by improving incoherence processing and applying vector quantization to incoherent weights. With additional fine-tuning, QuIP# achieves state-of-the-art performance in 2-bit models. However, when the bit-width is reduced below 4-bit, these methods all suffer from significant performance degradation compared to BF16 baselines. In contrast, pre-training quantization integrates quantization into the training process, requiring models to be trained from scratch, which results in better performance. Recent Ma et al. (2024) showed that ternary LLMs match the performance of full-precision counterpart starting from 3B parameter counts. Frantar & Alistarh (2024) quantized a 1.6 trillion parameter Switch Transformer to sub 1-bit precision.

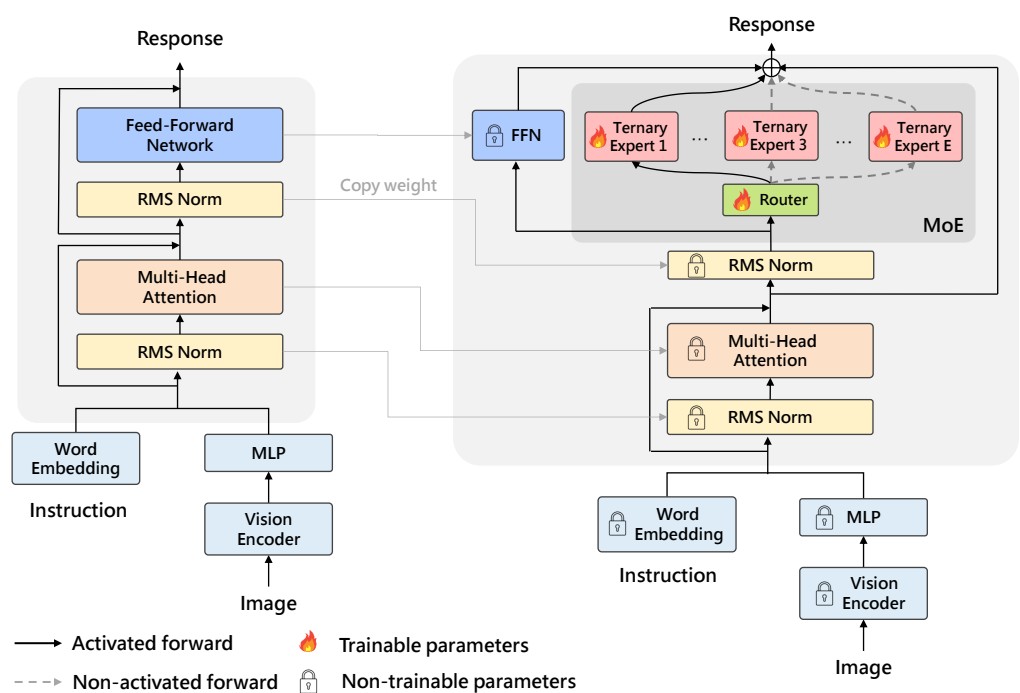

Figure 1: The overview of MoTE. We retain the pre-trained full-precision FFN as a shared expert and add a top-1 activated MoE layer with ternary experts. All experts and attention layers are initialized from the dense checkpoint.

Li et al. (2024b) proposed to quantize the experts with a mixed precision recipe and introduced a novel data-driven techniques for optimizing bit allocation.

## 3 MoTE: MIXTURE-OF-TERNARY-EXPERTS

### 3.1 ARCHITECTURE

We illustrate the architecture of MoTE in Figure 1. Previous studies (Komatsuzaki et al., 2023; Lin et al., 2024a) expanded a dense model into an MoE model by directly replacing the FFN layer with an MoE layer, where each expert is initialized from the dense FFN to accelerate convergence. However, as shown in Table 6, we found that directly replacing the FFN with an MoE in ternary up-cycling leads to significant performance degradation. We hypothesize that this occurs because the FFN encodes a substantial amount of factual knowledge acquired during pre-training (Geva et al., 2021; Dai et al., 2022), and weight ternarization severely disrupts pre-trained information. To mitigate this issue, we retain the FFN module from the dense model as a shared expert, ensuring it is activated for every token. Specifically, the forward computation of the $l$-th layer of MoTE can be formulated as:

$$x_l^a = x_{l-1} + \text{MSA}(\text{LN}(x_{l-1})) \tag{1}$$
$$x_l = x_l^a + \text{MoE}(\text{LN}(x_l^a)) + \text{FFN}(\text{LN}(x_l^a)) \tag{2}$$

where MSA and LN stands for multi-head self-attention and layer normalization, respectively. As illustrated in Figure 1, we initialize the FFN, MSA and MoE layers from the dense model. We implement the MoE mechanism following the GShard (Lepikhin et al., 2021), with each expert modeled as a Gated Linear Unit (GLU) (Shazeer, 2020). An MoE layer which consists of $E$ ternary experts $\text{FFN}_1^T$ ... $\text{FFN}_E^T$ satisfies that:

$$\text{MoE}(x) = \sum_{i=1}^{E} \mathcal{P}(x)_i \cdot \text{FFN}_i^T(x), \quad \mathcal{P}(x)_i = \frac{e^{f(x)_i}}{\sum_{j=1}^{E} e^{f(x)_j}} \tag{3}$$

where $f(x)$ is the gating logits produced by the router. We leave the projection in router as BF16, since it only accounts for very small portion of total memory footprint. The forward computation of the $i$-th ternary expert $\mathrm{FFN}_i^T(x)$ satisfies that:

$$\mathrm{FFN}_i^T(x) = Q_w(W_{\mathrm{down}}^T)Q_a(h) \tag{4}$$

$$h = Q_w(W_{\mathrm{up}}^T)Q_a(x) \otimes \sigma[Q_w(W_{\mathrm{gate}}^T)Q_a(x)] \tag{5}$$

$\sigma$ is SiLU function. We apply *absmean* quantizer and *per-token absmax* quantizer for weight and activation quantization in expert's linear layers following BitNet (Ma et al., 2024). Specifically, the quantization can be formulated as:

$$Q_w(W) = \alpha \cdot \mathrm{RoundClip}(\frac{W}{\alpha}, -1, 1), \tag{6}$$

$$Q_a(x) = \frac{\beta}{127} \cdot \mathrm{RoundClip}(\frac{127x}{\beta}, -128, 127) \tag{7}$$

$$\alpha = \frac{1}{nm}||W||_1, \quad \beta = ||x||_\infty \tag{8}$$

$$\mathrm{RoundClip}(x, a, b) = \max(a, \min(b, \mathrm{round}(x))) \tag{9}$$

The weight $W \in \mathcal{R}^{m \times n}$ is quantized into ternary values, i.e., $\{-1, 0, 1\}$. The activations $x$ are per-token quantized into 8-bit integers, i.e., $[-128, 127]$. The output of ternary linear layer $Y$ is $Q_w(W)Q_a(x)$. During inference, we use the kernel from BitBlas (Wang et al., 2024c) to save the memory footprint and accelerate the inference. Despite ternary values results in 1.58-bit, i.e., $\log 3 / \log 2$, BitBlas still stores and processes ternary weight in INT2 format since current GPUs are still based on binary system.

## 3.2 TRAINING RECIPE

Following MoE-LLaVA (Lin et al., 2024a), the training of MoTE consists of three stages. In Stage I, we train a two-layer MLP connector to align the visual encoder and LLM. As for Stage II, we fine-tune the LLM and connector using more complex vision-language instruction data. In Stage III, we expand the dense model from Stage II to an MoE model with ternary experts. The visual encoder is frozen through the training process. As presented in Figure 1, during up-cycling, only ternary MoE layers are trainable, and the shared expert and MSA layers are frozen.

We adopt quantization-aware training for MoTE. The weights and activations are quantized into ternary and INT8 values on-the-fly. Since many operations in the quantization are no-differentiable, we deploy straight-through estimator (Bengio et al., 2013) for gradient approximation. The gradients are directly by-passing through non-differentiable functions, i.e., $\frac{\partial \mathcal{L}}{\partial W} = \frac{\partial \mathcal{L}}{\partial Q(W)}$ and $\frac{\partial \mathcal{L}}{\partial X} = \frac{\partial \mathcal{L}}{\partial Q(X)}$. The gradients and optimizer states are retained as full-precision.

## 3.3 TRAINING OBJECTIVES

The training objective of MoTE $\mathcal{L}_{\mathrm{total}}$ requires the minimization of both the loss of specific multimodal tasks $\mathcal{L}_{\mathrm{LM}}$ and an auxiliary load balancing loss $\mathcal{L}_{\mathrm{balance}}$.

**Language modeling loss.** The auto-regressive language modeling loss $\mathcal{L}_{\mathrm{LM}}$ is widely adopted in the training of LMMs. Specifically, let $\mathcal{V}$ and $\mathcal{T}$ denote sequences of visual tokens and textual tokens, respectively. $\mathcal{T}$ can be divided as the instruction part $\mathcal{T}_{ins}$ and the response part $\mathcal{T}_{ans}$. The language modeling loss is calculated as:

$$\mathcal{L}_{\mathrm{LM}} = - \sum_{\mathrm{token}_i \in \mathcal{T}_{ans}} \log \Pr(\mathcal{Y}^i \,|\, \mathcal{V}, \mathcal{T}^{[:i-1]}) \tag{10}$$

where $\mathcal{Y}$ is the model's output. We only calculate the loss on the response part.

**Load balancing loss.** To ease the expert load imbalance problem in MoE, we adopt an auxiliary loss following Switch Transformers (Fedus et al., 2022). Given a batch of training tokens $\mathbf{X}$, the

balancing loss can be formulated as:

$$\mathcal{L}_{\text{balance}} = \frac{E}{|\mathbf{X}|} \sum_{i=1}^{E} \sum_{x \in \mathbf{X}} t_i \cdot \mathcal{P}(x)_i \tag{11}$$

where $|\mathbf{X}|$ is the number of training tokens in $\mathbf{X}$, $\mathcal{P}(x)_i$ is the routing logits depicted in Equation 3, $t_i$ is the number of tokens routed to the $i$-th expert.

Above all, the training objective of MoTE is:

$$\mathcal{L}_{\text{total}} = \mathcal{L}_{\text{LM}} + \gamma \cdot \mathcal{L}_{\text{balance}} \tag{12}$$

where $\gamma$ is a coefficient for load balancing.

## 4 EXPERIMENTS

### 4.1 SETUP

**Model settings.** We select MoE-LLaVA (Lin et al., 2024a) as the baseline. It adopts a similar three-stage MoE training recipe and utilizes full-precision experts. Since MoE-LLaVA activates the top-2 experts, and our model includes a shared expert, we use top-1 gating in MoTE to ensure a fair comparison in terms of FLOPs. All MoE layers consist of four routed experts. We adopt SigLIP-L (Zhai et al., 2023) as the vision encoder and the instruct-version of Qwen2.5-series model (Yang et al., 2024) as the base LLM. The connector is a two-layer MLP with GELU activation. Table 1 presents the active and total parameter counts in

Table 1: The active/total parameter counts and expert memory of MoTE and MoE-LLaVA in various model sizes.

| Method | # Active/Total Params | | | Expert Memory ↓ |
|---|---|---|---|---|
| | Stage I | Stage II | Stage III | |
| *0.5B Model Up-cycling* | | | | |
| MoE-LLaVA | 1B | 1B | 1.3B/1.8B | 2.3GB (2.55×) |
| MoTE | | | 1.3B/2.1B | **0.9GB (1.00×)** |
| *1.5B Model Up-cycling* | | | | |
| MoE-LLaVA | 2B | 2B | 3.1B/5.4B | 8.6GB (2.69×) |
| MoTE | | | 3.1B/6.6B | **3.2GB (1.00×)** |
| *3B Model Up-cycling* | | | | |
| MoE-LLaVA | 3.4B | 3.4B | 5.9B/10.8B | 18.1GB (2.66×) |
| MoTE | | | 5.9B/13.2B | **6.8GB (1.00×)** |

the training of MoTE and MoE-LLaVA across different model sizes. The expert memory footprint includes contributions from both shared and routed experts.

**Implementation details.** We adopt expert parallelism for efficient training of MoE models. The coefficient $\gamma$ for load balancing loss is set as 0.01. The value is recommended by Fedus et al. (2022) to ensure auxiliary loss not to overwhelm the primary language modeling objective. All experiments are conducted on 16 NVIDIA A100 cards with 40GB memory. Due to the limited computation resources, we do not perform dynamic resolution processing for the images, since it leads to extremely long training sequence. The length of the total sequence is set as 2048 tokens, and the visual input includes 729 tokens. More hyper-parameters can be found in Appendix A.

**Training data.** We train MoTE and MoE-LLaVA on the same dataset to ensure a fair comparison. The training dataset consists of a total of 5 million samples. For the first stage, we use the pre-training data of LLaVA 1.5 (Liu et al., 2024a). For the second stage, we use the mixture of SViT (Zhao et al., 2023), LVIS (Wang et al., 2023), LRV (Liu et al., 2023) and MIMIC-IT (Li et al., 2023a). For the third stage, we use a subset of MAmmoTH-VL (Guo et al., 2024), which includes 3.4 million instruction-response pairs, each associated with a single image as the visual input.

**Evaluation.** We report the zero-shot performance of these models on a range of image under-standing tasks using LMM-Eval toolkit (Zhang et al., 2024b), including MMMU (Yue et al., 2024), MathVista (Lu et al., 2024) (MathV), MMBench (Liu et al., 2024b) (MMB), MMStar (Chen et al., 2024a) (MMS), MM-Vet (Yu et al., 2023) (MMV), SeedBench-2-Plus (Li et al., 2024a) (Seed$^{2+}$), SeedBench (Li et al., 2023b) (Seed), AI2D (Kembhavi et al., 2016), ChartQA (Masry et al., 2022), InfoVQA (Mathew et al., 2022) and DocVQA (Mathew et al., 2021).

Table 2: The results of MoTE and MoE-LLaVA on image understanding tasks in different model sizes. All models utilize the same base LLM, vision encoder and training dataset to ensure a fair comparison.

| Method | MMMU (val) | MathV (testmini) | MMB (en test) | MMS (test) | Seed$^{2+}$ (test) | AI2D (test) | ChartQA (test) | InfoVQA (val) | DocVQA (val) | Avg. |
|---|---|---|---|---|---|---|---|---|---|---|
| *0.5B Model Up-cycling* | | | | | | | | | | |
| MoE-LLaVA | 35.4 | 35.4 | 57.3 | 39.5 | 43.3 | 57.4 | 56.0 | 25.8 | 49.3 | 44.4 |
| **MoTE** | 34.2 | 35.2 | 57.6 | 37.9 | 44.8 | 55.2 | 54.9 | 25.2 | 49.7 | 43.8 |
| △ *compare to MoE-LLaVA* | -1.2 | -0.2 | +0.3 | -1.6 | +1.5 | -2.2 | -1.1 | -0.6 | +0.4 | -0.6 |
| *1.5B Model Up-cycling* | | | | | | | | | | |
| MoE-LLaVA | 41.2 | 41.7 | 68.4 | 45.0 | 52.9 | 67.8 | 59.4 | 31.8 | 55.1 | 51.5 |
| **MoTE** | 42.6 | 44.8 | 70.0 | 46.4 | 54.8 | 68.7 | 61.3 | 32.5 | 57.4 | 53.2 |
| △ *compare to MoE-LLaVA* | +1.4 | +3.1 | +1.6 | +1.4 | +1.9 | +0.9 | +1.9 | +0.7 | +2.3 | +1.7 |
| *3B Model Up-cycling* | | | | | | | | | | |
| MoE-LLaVA | 42.3 | 48.6 | 75.4 | 45.5 | 56.2 | 73.5 | 65.0 | 35.1 | 60.1 | 55.7 |
| **MoTE** | 43.4 | 52.3 | 74.5 | 48.2 | 57.5 | 73.9 | 67.6 | 36.7 | 61.3 | 57.3 |
| △ *compare to MoE-LLaVA* | +1.1 | +3.7 | -0.9 | +2.7 | +1.3 | +0.4 | +2.6 | +1.6 | +1.2 | +1.6 |

Table 3: The results of MoTE and MoE-LLaVA given the same amount of expert memory in 1.5B and 3B model size. Both of them are combined with post-training quantization (PTQ). The expert memory footprint includes contributions from both shared and routed experts.

| Method | Expert Memory↓ | MMMU↑ (val) | MMB↑ (en test) | Seed$^{2+}$↑ (test) | AI2D↑ (test) | DocVQA↑ (val) | Avg.↑ |
|---|---|---|---|---|---|---|---|
| *1.5B Model Up-cycling* | | | | | | | |
| MoE-LLaVA + PTQ | 2.2GB | 41.1 | 68.0 | 53.1 | 67.3 | 55.0 | 56.9 |
| MoTE + PTQ | 2.2GB | 42.7 | 70.1 | 54.4 | 68.2 | 57.4 | 58.6 |
| MoE-LLaVA + PTQ | 1.6GB | 36.0 | 60.3 | 49.8 | 62.6 | 50.0 | 51.7 |
| MoTE + PTQ | 1.6GB | 40.3 | 69.3 | 55.2 | 67.8 | 57.1 | 57.9 |
| *3B Model Up-cycling* | | | | | | | |
| MoE-LLaVA + PTQ | 4.5GB | 42.2 | 75.3 | 55.4 | 72.3 | 59.4 | 60.9 |
| MoTE + PTQ | 4.5GB | 43.2 | 74.8 | 57.0 | 73.3 | 60.9 | 61.8 |
| MoE-LLaVA + PTQ | 3.4GB | 37.7 | 69.7 | 52.2 | 67.5 | 56.8 | 56.8 |
| MoTE + PTQ | 3.4GB | 42.8 | 71.9 | 56.9 | 73.0 | 60.9 | 61.1 |

## 4.2 MAIN RESULTS

We compared the performance of ternary up-cycling MoTE to MoE-LLaVA across different model sizes on various multimodal tasks. As shown in Table 2, MoTE underperformed full-precision up-cycling MoE-LLaVA when converting a 0.5B dense model to an MoE model. However, the performance gap between MoTE and MoE-LLaVA narrows as the parameter counts of the dense model increases. Similar phenomenons are also reported by the low-bit pre-training of LLMs (Ma et al., 2024; Kaushal et al., 2024), which suggests promising trends of scaling model size for ternary MoEs.

As the model size scales to 1.5B parameters, due to larger total parameter counts, MoTE surpasses MoE-LLaVA across various image understanding tasks, achieving an average accuracy improvement of 1.7% with the same FLOPs. This demonstrates the effectiveness of our proposed method. Moreover, since the expert weights in MoTE are trained to adapt to ternary values, despite it has larger total parameter counts, the ternary MoE layer can be losslessly compressed to low-bit after training, significantly reducing the memory footprint caused by the ensemble of experts. As shown in Table 1, at the 3B model size, MoTE's expert memory is only 6.8GB — just 38% of MoE-LLaVA's 18.1GB.

## 4.3 COMPATIBILITY WITH POST-TRAINING QUANTIZATION

Despite the MoE layers of our model contain ternary experts, there still leaves a shared expert in full-precision in each layer. These shared experts can be quantized into low-bit using post-training quantization methods. We apply GPTQ (Frantar et al., 2022) and AWQ (Lin et al., 2024b) at various bit-widths and report the best results given the same expert memory footprint. We use 512 samples

Table 4: The results of MoTE and the other methods in similar model size on general VQA and multimodal reasoning tasks.

| Model | Training Tokens | MMMU (val) | MMB (en test) | Seed (image) | MMS (test) | MMV (test) | MathV (testmini) | Avg.↑ |
|---|---|---|---|---|---|---|---|---|
| *Dense Model* | | | | | | | | |
| MM1.5-1B (Zhang et al., 2024a) | >200B | 35.8 | - | 70.2 | - | 37.4 | 37.2 | - |
| MM1.5-3B (Zhang et al., 2024a) | >200B | 37.1 | - | 72.4 | - | 41.0 | 44.4 | - |
| MiniCPM-V2-3B (Yao et al., 2024) | - | 38.2 | 69.1 | - | 41.7 | - | 38.7 | - |
| TinyLLaVA-3B (Zhou et al., 2024) | 4B | 39.9 | - | - | - | 34.8 | - | - |
| Phi-3-Vision-4B (Abdin et al., 2024) | >0.8T | 40.4 | 73.9 | 71.8 | 47.9 | 45.4 | 44.5 | 54.0 |
| Qwen2-VL-2B (Wang et al., 2024d) | >1.4T | 41.1 | 74.9 | 72.1 | 48.0 | 49.5 | 43.0 | 54.8 |
| *Sparse Model* | | | | | | | | |
| MoE-LLaVA (Lin et al., 2024a) | 4B | 33.9 | 52.6 | 64.8 | 32.5 | 32.3 | 25.6 | 40.3 |
| MolmoE-1B (Deitke et al., 2024) | 1.5B | 34.9 | 63.6 | 68.7 | 43.3 | 38.5 | 34.0 | 47.2 |
| LLaVA-MoD-2B (Shu et al., 2024) | 10B | - | 68.9 | - | - | - | - | - |
| MM1-3B-MoE (McKinzie et al., 2024) | >400B | 38.6 | 70.8 | 69.4 | - | 42.2 | 32.6 | - |
| MM1-7B-MoE (McKinzie et al., 2024) | >400B | 40.9 | 72.7 | 70.9 | - | 45.2 | 40.9 | - |
| MM1.5-1B-MoE (Zhang et al., 2024a) | >200B | 41.2 | - | 71.4 | - | 39.8 | 42.9 | - |
| **MoTE-1.5B (ours)** | 21.6B | 40.4 | **75.0** | 72.5 | 50.2 | 52.6 | **49.8** | **56.8** |
| w/o initialize experts from FFN | 21.6B | **41.8** | **75.0** | 71.3 | 48.1 | 48.6 | 48.2 | 55.5 |

Table 5: Ablations on the precision of routed experts in MoTE.

| Precision of Routed Expert | MMMU (val) | MMB (en test) | AI2D (test) | ChartQA (test) | Seed[2+] (test) | MMS (test) | Avg.↑ |
|---|---|---|---|---|---|---|---|
| 1-bit | 40.3 | 69.5 | 67.6 | 60.2 | 53.9 | 43.1 | 55.7 |
| **1.58-bit** | **42.6** | **70.0** | **68.7** | **61.3** | **54.8** | **46.4** | **57.3** |

with the length of 2048 tokens from Stage III's data as the calibration set. For MoE-LLaVA, all full-precision experts are quantized, resulting in expert memory footprints of 2.2GB and 4.5GB under INT4 quantization for the 1.5B and 3B models, respectively. To ensure a fair comparison, we quantize the shared expert of MoTE to INT8 using RTN (Dettmers et al., 2022). Additionally, we extend the comparison to scenarios with lower memory constraints. For expert memory footprints of 1.6GB and 3.4GB in the 1.5B and 3B models, MoE-LLaVA's experts are quantized to 3-bit integers using GPTQ, while the shared experts of MoTE are quantized to INT4.

Table 3 presents the results for MoTE and MoE-LLaVA, both combined with post-training quantization. Given the same expert memory footprint, MoTE achieves better performance than MoE-LLaVA. Under the same expert memory footprint, our method outperforms MoE-LLaVA across different model sizes. Notably, under stricter memory constraints, we observe a significant performance drop for MoE-LLaVA combined with GPTQ at 3-bit precision. However, since the parameters of our MoE layer are ternary, we can achieve the same memory footprint by applying INT4 quantization only to the shared expert. This further amplifies the advantages of our approach. Specifically, given the same expert memory of 3.4GB, MoTE achieves a gain of 4.3% average accuracy compared with MoE-LLaVA on the end tasks. These results demonstrate that our method can achieve lower memory footprint combined with post-training quantization, while maintaining competitive performance.

### 4.4 SCALING WITH MORE DATA

To examine whether our method is friendly for scaling with data, we train a 1.5B MoTE model with more data during ternary up-cycling. We adopt the same data recipe for Stage I and Stage II as shown in Section 4.1. Then we use a full set of MammoTH-VL (Guo et al., 2024) for Stage III, which contains 10 million samples, each associated with a single image. Every dense layer is replaced with an MoTE layer with one full-precision shared expert and four routed ternary experts. The training steps is set as 40k. The other hyper-parameters are consistent with the setup presented in Section 4.1.

Table 4 summarizes the zero-shot accuracy of MoTE and the baselines across various multimodal reasoning and general VQA tasks. For the baselines, we use their reported scores when available; otherwise, we evaluate the open-sourced models using the same prompts as ours to ensure a fair

Table 6: Ablations on the precision of shared experts and the initialization methods of routed experts in MoTE.

| Precision of Shared Expert | Initialize from FFN | MMMU (val) | MMB (en test) | AI2D (test) | ChartQA (test) | Seed$^{2+}$ (test) | MMS (test) | Avg.↑ |
|---|---|---|---|---|---|---|---|---|
| Ternary | ✗ | 34.6 | 49.4 | 62.7 | 56.4 | 46.2 | 39.8 | 48.2 |
| BF16 | ✗ | 40.1 | 69.9 | 67.1 | 59.9 | 53.2 | 44.5 | 55.8 |
| **BF16** | ✓ | **42.6** | **70.0** | **68.7** | **61.3** | **54.8** | **46.4** | **57.3** |

Table 7: Ablations on the training recipe of MoTE. Given the same training FLOPs, we do not observe performance improvement from initially training with full-precision experts then fine-tuning them into ternary precision.

| Ternary Training | Full-Precision Training | MMMU (val) | MMB (en test) | AI2D (test) | ChartQA (test) | Seed$^{2+}$ (test) | MMS (test) | Avg.↑ |
|---|---|---|---|---|---|---|---|---|
| 20% | 80% | 39.3 | 60.5 | 62.6 | 56.8 | 53.2 | 42.0 | 52.4 |
| 60% | 40% | 41.3 | 64.0 | 65.3 | 57.0 | 54.0 | 45.1 | 54.4 |
| **100%** | **0%** | **42.6** | **70.0** | **68.7** | **61.3** | **54.8** | **46.4** | **57.3** |

comparison. As shown in Table 4, although MoTE-1.5B is only trained with 21.6B tokens, our model achieves an improvement of 2.0% average accuracy compared to Qwen2-VL-2B (Wang et al., 2024d). Furthermore, MoTE outperforms the larger dense model with fewer FLOPs. Specifically, MoTE outperforms MiniCPM-V-2.0-3B and Phi-3-Vision-4B by a gain of 11.1% and 5.3% accuracy on the *testmini* set of MathVista.

For sparse model, due to stronger base LLM and vision encoder, our model significantly outperforms MoE-LLaVA of similar total and active model size by a gain of 16.5% average accuracy. Notably, MM1.5-1B-MoE is a strong multimodal MoE baseline, which was trained from an 1B dense model with 64 experts replacing dense layers every two layers. MoTE outperforms it by a gain of 0.6%, 1.1%, 12.8% and 6.9% on MMMU, SeedBench (image), MMVet and MathVista, respectively. These results proves the effectiveness of the proposed MoTE on multimodal reasoning and general VQA.

## 4.5 ABLATION STUDIES

**Precision of routed experts.** We investigate the impact of expert precision on the performance of MoTE. Specifically, we compare ternary (i.e., 1.58-bit) up-cycling to 1-bit up-cycling with BWN (Rastegari et al., 2016) as the weight quantizers. Both models are up-cycled from Qwen2.5-1.5B with SigLIP-L as the vision encoder to ensure a fair comparison. As shown in Table 5, using binary experts results in performance degradation across most tasks. Similar findings have been reported in the quantization-aware training of BERT models (Bai et al., 2021), where transitioning from ternary to binary weights leads to a substantially more complex and irregular loss landscape, making optimization notably more difficult. Above all, ternary up-cycling is a memory-effective and high-performance solution for MoE models.

**Precision of shared experts.** We ablate the effect of the precision of the shared expert reused from the FFN of pre-trained dense checkpoint. MoTE retains the precision of shared expert as BF16 and freezes the modules during up-cycling. We compare it to a model with the ternary shared expert. All ternary experts are trainable. Table 6 presents the zero-shot performance of these models on MMMU, MMBench, AI2D, ChartQA, SeedBench-2-Plus and MMStar tasks. Weight ternarization of the shared experts has significant effect on overall performance. Specifically, the model with full-precison shared experts outperforms it with ternary shared experts by an improvement of 7.6% average accuracy on the end tasks. This demonstrates the importance of keeping the pre-trained FFN as a high-precision shared expert during ternary up-cycling.

**Initialization of routed experts.** We compare MoTE to randomly initialized routed experts in Stage III. Table 6 presents the results for a 1.5B model, where initializing from the FFN yields a 1.5% improvement in average accuracy on end tasks compared to random initialization. Moreover, we

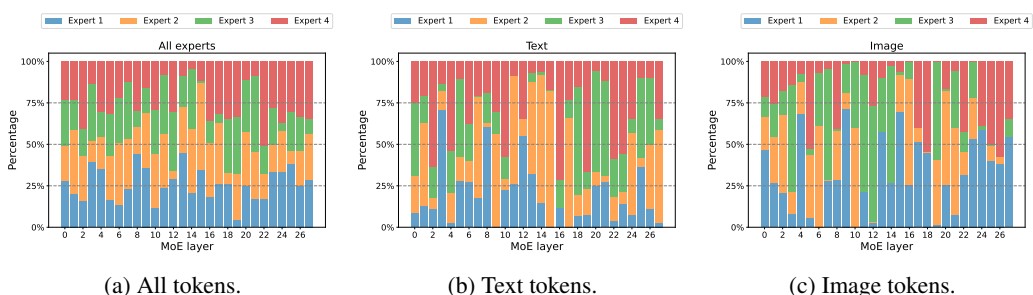

(a) All tokens.                    (b) Text tokens.                    (c) Image tokens.

Figure 2: Visualization of the routing distributions of all tokens, text tokens, image tokens across all experts on the *en-test* set of MMBench.

analyze the impact of data scaling using the data recipe described in Section 4.4. As demonstrated in Table 4, FFN-based initialization maintains its advantage with additional training data, achieving a 1.3% higher average accuracy than random initialization. These findings suggest that leveraging a pre-trained full-precision FFN for MoTE's initialization not only enhances performance but also accelerates the convergence of ternary experts. Additional results for the 0.5B and 3B models are provided in the Appendix B.

**Training recipe.** We conduct ablation studies on the training strategy of ternary up-cycling in MoTE to assess the effectiveness of first training with full-precision experts before fine-tuning the model to ternary precision. All models are trained on 6.25B tokens and up-cycled from Qwen2.5-1.5B. We vary the proportion of training conducted in full-precision versus ternary precision. As shown in Table 7, we do not observe performance gain from initially training with full-precision experts. In fact, accuracy improves as the proportion of ternary training increases. Therefore, for both simplicity and improved performance, MoTE is trained directly in ternary precision without a full-precision training phase during up-cycling.

## 5 ANALYSIS

We visualize the routing distribution of all tokens in MoTE-1.5B on the *en-test* split of the MMBench dataset. As shown in Figure 2a, expert utilization across all tokens is well-balanced. To further investigate modality-specific behavior, we present the routing distributions for text and image tokens separately in Figures 2b and 2c, respectively. Notably, text and image tokens exhibit distinct routing patterns. For example, expert #1 is frequently activated for image tokens in the first layer and the final five layers. Additional visualizations across various tasks are provided in Appendix C.1. We observe that routing distributions remain largely consistent across different tasks, suggesting that the experts in MoTE specialize based on modality rather than task-specific features. Moreover, we include per-expert routing distributions by modality in Appendix C.2. Interestingly, some experts exhibit clear modality preferences despite the absence of explicit modality conditioning during training. To better understand expert specialization, we further apply PCA to extract the top-10 routing pathways for text and image tokens. More visualizations are included in Appendix C.3. These findings enhance our understanding of MoTE's behavior and workflow from a token-level perspective.

## 6 CONCLUSION

In this work, we introduce MoTE, a scalable and memory-efficient approach to train multimodal Mixture-of-Ternary-Experts models from full-precision dense checkpoints. Extensive experiments show that our model matches the full-precision up-cycling MoE-LLaVA in zero-shot performance on end tasks, starting from model sizes exceeding 1.5B parameters. Furthermore, MoTE is compatible with post-training quantization methods, enabling further reductions in the memory footprint of MoE models. Given the same expert memory footprint of 3.4GB, MoTE surpasses MoE-LLaVA with an average accuracy gain of 4.3% on image understanding tasks, highlighting the effectiveness of our approach, particularly for memory-constrained edge devices.

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

## A    HYPER-PARAMETERS

In this section, we present the detailed hyper-parameters used for the training of MoTE and full-precision up-cycling baseline MoE-LLaVA. For Stage I and Stage II, we adopt the same training recipe, data and hyper-parameters, for both MoTE and MoE-LLaVA. For Stage III, we use the learning rate and scheduler recommended by MoE-LLaVA for full-precision training. For MoTE, following BitNet, we use a much large learning rate and two-stage weight decay for ternary experts which is critical for the optimization of extremely low-bit training.

We utilize *torch.compile* to compile the PyTorch code in the quantization into optimized kernels, which significantly speed up the training of MoTE. As for the training of 1.5B model's up-cycling in Stage III, MoTE costs 43.3 hours on 16 NVIDIA A100 cards (40GB), while MoE-LLaVA uses 41.8 hours. Above all, MoTE has similar training time compared to full-precision up-cycling MoE-LLaVA.

Table 8: Hyper-parameters for the training of MoTE and MoE-LLaVA with 0.5B model. $a/b$ denotes the value of MoTE/MoE-LLaVA. $1 + 4$ denotes that the model has one shared expert and four routed experts.

| Hyper-parameter | Stage I | Stage II | Stage III |
|---|---|---|---|
| Learning rate | 1e-3 | 5e-5 | 1.5e-4/5e-5 |
| Batch Size | 256 | 128 | 256 |
| Weight decay | ✗ | ✗ | $0.1 \rightarrow 0$/✗ |
| Training steps | 2500 | 8000 | 12500 |
| Training sequence | 1024 | 1024 | 2048 |
| Vision sequence | | 729 | |
| AdamW $\beta$ | | (0.9, 0.999) | |
| AdamW $\epsilon$ | | 1e-8 | |
| # MoE layer | - | - | 24 |
| # Experts | - | - | 1+4 / 0+4 |
| # Top-$k$ | - | - | 1+1 / 0+2 |

Table 9: Hyper-parameters for the training of MoTE and MoE-LLaVA with 1.5B and 3B model. $a/b$ denotes the value of MoTE/MoE-LLaVA. $1 + 4$ denotes that the model has one shared expert and four routed experts.

| Hyper-parameter | Stage I | Stage II | Stage III |
|---|---|---|---|
| Learning rate | 1e-3 | 2e-5 | 1e-4/2e-5 |
| Batch Size | 256 | 128 | 256 |
| Weight decay | ✗ | ✗ | $0.1 \rightarrow 0$/✗ |
| Training steps | 2500 | 8000 | 12500 |
| Training sequence | 1024 | 1024 | 2048 |
| Vision sequence | | 729 | |
| AdamW $\beta$ | | (0.9, 0.999) | |
| AdamW $\epsilon$ | | 1e-8 | |
| # MoE layer | - | - | 28 |
| # Experts | - | - | 1+4 / 0+4 |
| # Top-$k$ | - | - | 1+1 / 0+2 |

## B  MORE ABLATION STUDIES

We compare MoTE with the randomly initialized routed experts in Stage III. We evaluate the zero-shot performance of these models on a range of image understanding tasks, including MMMU, MMBench, AI2D, ChartQA, SeedBench-2-Plus and MMStar dataset.

Table 10 shows the results of both methods in 0.5B, 1.5B and 3B model size. Initializing from FFN outperforms random initialization by a gain of 1.0%, 1.5% and 0.3% average accuracy on end tasks in 0.5B, 1.5B and 3B model size, respectively. The results demonstrate that using the pre-trained full-precision FFN for MoTE's initialization achieves better performance across various model size.

Table 10: Ablations on the initialization methods of the routed experts for MoTE across different model sizes.

| Initialize from FFN | MMMU | MMBench | AI2D | ChartQA | SeedBench$^{2+}$ | MMStar | Avg. |
|---|---|---|---|---|---|---|---|
| *0.5B Model Up-cycling* | | | | | | | |
| ✗ | **34.8** | 50.5 | **55.2** | **55.8** | 43.0 | **39.1** | 46.4 |
| ✓ | 34.2 | **57.6** | **55.2** | 54.9 | **44.8** | 37.9 | **47.4** |
| *1.5B Model Up-cycling* | | | | | | | |
| ✗ | 40.1 | 69.9 | 67.1 | 59.9 | 53.2 | 44.5 | 55.8 |
| ✓ | **42.6** | **70.0** | **68.7** | **61.3** | **54.8** | **46.4** | **57.3** |
| *3B Model Up-cycling* | | | | | | | |
| ✗ | 43.3 | **75.5** | 72.7 | 65.5 | 57.1 | **48.8** | 60.5 |
| ✓ | **43.4** | 74.5 | **73.9** | **67.6** | **57.5** | 48.2 | **60.8** |

## C  VISUALIZATION

We visualize the workflows of MoTE-1.5B at three distinct levels: expert, modality, and token. Specifically, we selected the AI2D, SeedBench-2-Plus, ChartQA, DocVQA, InfoVQA, MMStar, and MMBench datasets. Figures 3, 4, and 5 respectively illustrate the load distributions across different experts, the modality-aware routing distributions for each expert, and the top-10 activated pathways obtained via PCA. Our analysis indicates that, although the routing distributions of MoTE remain quite similar across tasks, they are predominantly influenced by the input modality.

### C.1  ROUTING DISTRIBUTION FOR TOKENS

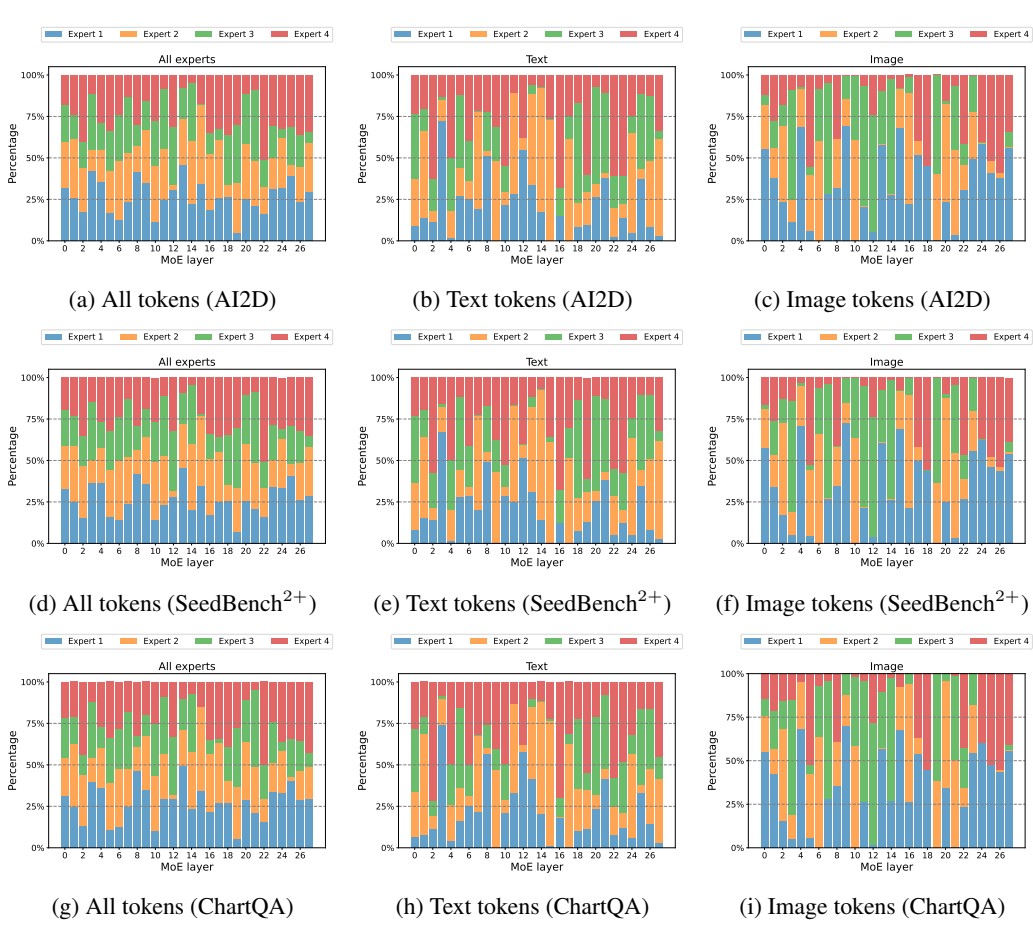

(a) All tokens (AI2D)  (b) Text tokens (AI2D)  (c) Image tokens (AI2D)

(d) All tokens (SeedBench$^{2+}$)  (e) Text tokens (SeedBench$^{2+}$)  (f) Image tokens (SeedBench$^{2+}$)

(g) All tokens (ChartQA)  (h) Text tokens (ChartQA)  (i) Image tokens (ChartQA)

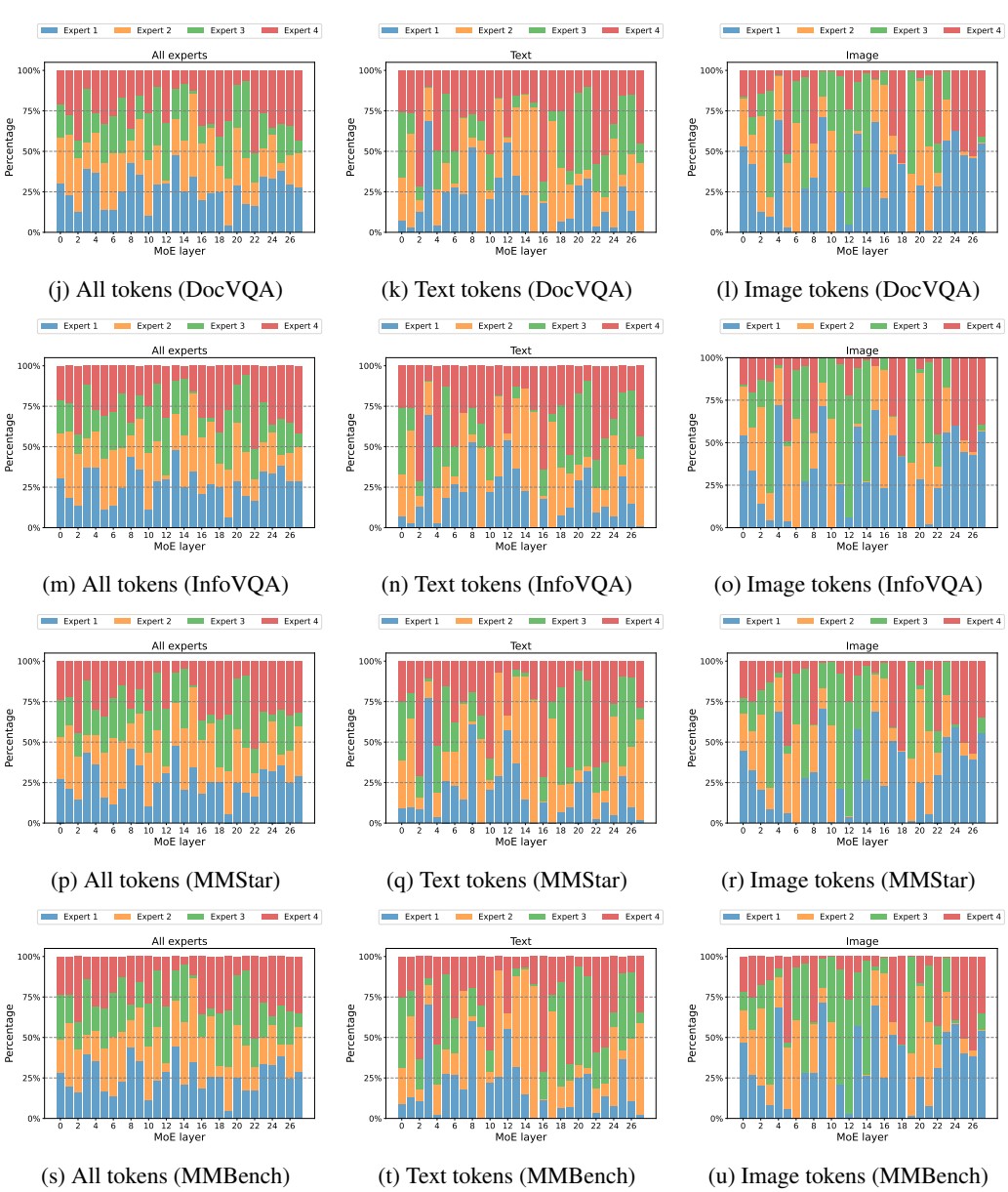

(j) All tokens (DocVQA)  (k) Text tokens (DocVQA)  (l) Image tokens (DocVQA)

(m) All tokens (InfoVQA)  (n) Text tokens (InfoVQA)  (o) Image tokens (InfoVQA)

(p) All tokens (MMStar)  (q) Text tokens (MMStar)  (r) Image tokens (MMStar)

(s) All tokens (MMBench)  (t) Text tokens (MMBench)  (u) Image tokens (MMBench)

Figure 3: Visualization of the routing distributions of all tokens, text tokens, image tokens across all experts on various tasks.

## C.2 ROUTING DISTRIBUTION FOR EACH EXPERTS

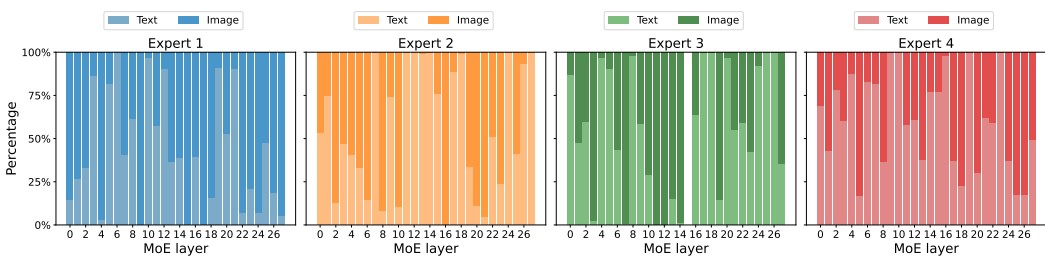

(a) Routing distribution on AI2D.

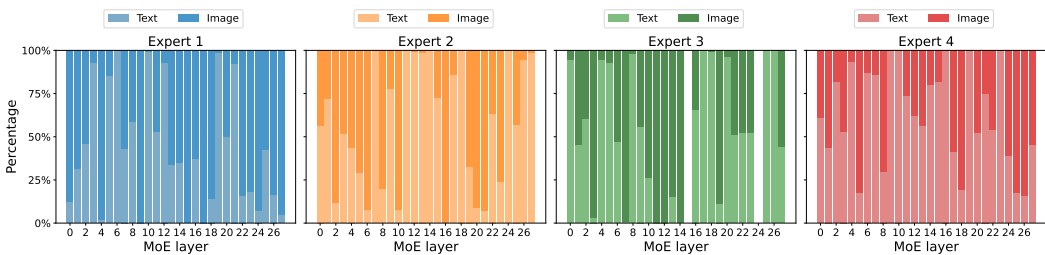

(b) Routing distribution on SeedBench-2-Plus.

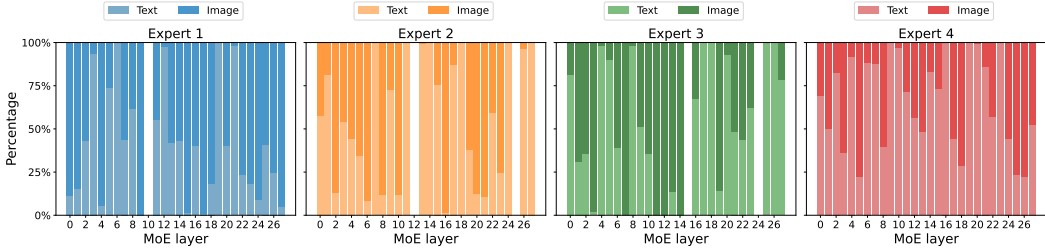

(c) Routing distribution on ChartQA.

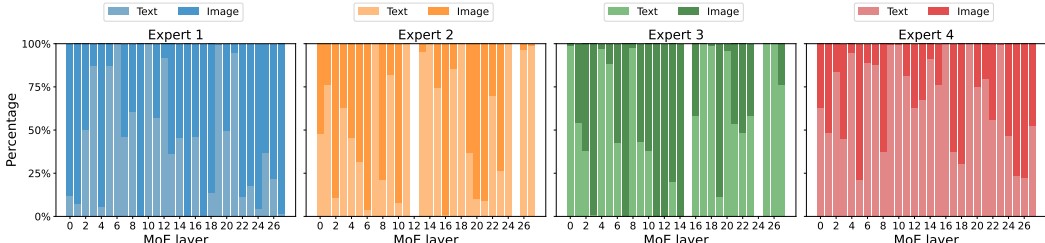

(d) Routing distribution on DocVQA.

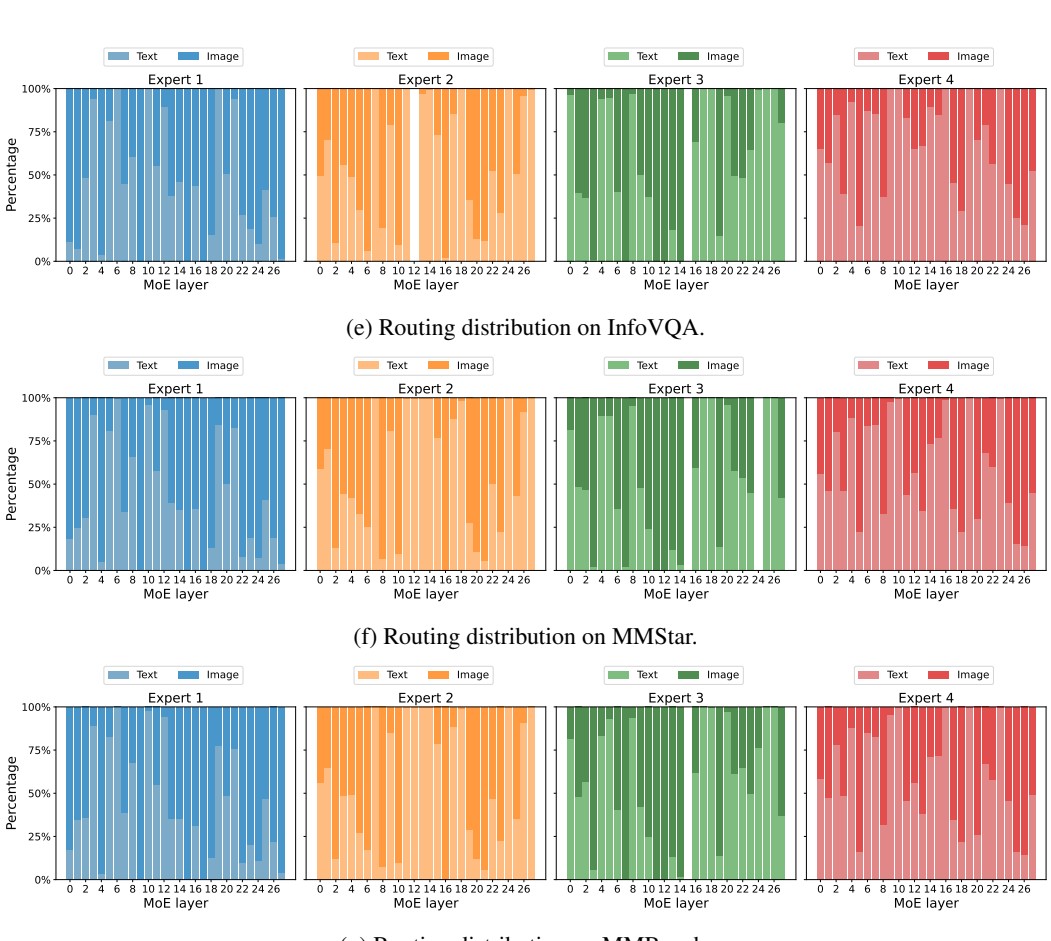

(e) Routing distribution on InfoVQA.

(f) Routing distribution on MMStar.

(g) Routing distribution on MMBench.

Figure 4: Visualization of the modality-aware routing distributions for each expert on various tasks.

## C.3 ACTIVATED PATHWAYS

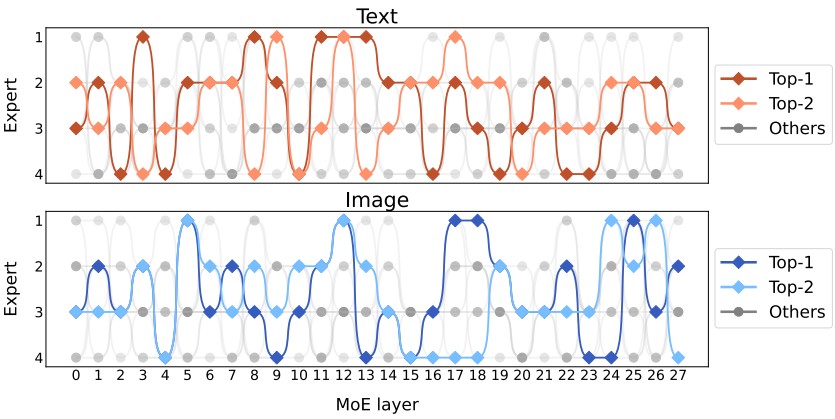

(a) The top-10 pathways for text and image tokens on MMBench.

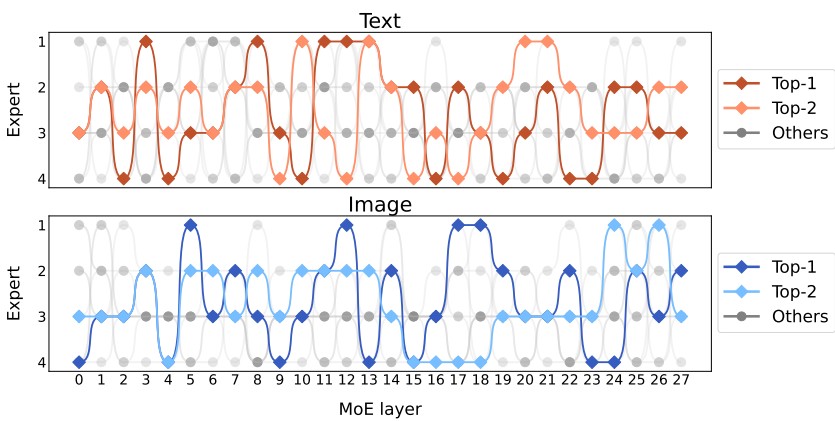

(b) The top-10 pathways for text and image tokens on AI2D.

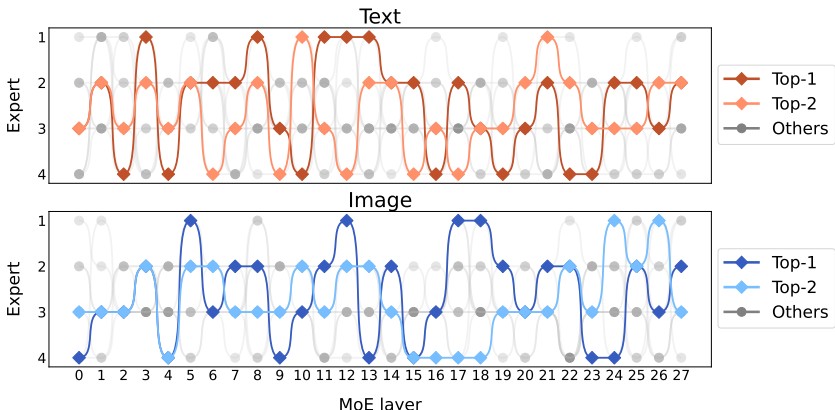

(c) The top-10 pathways for text and image tokens on SeedBench-2-Plus.

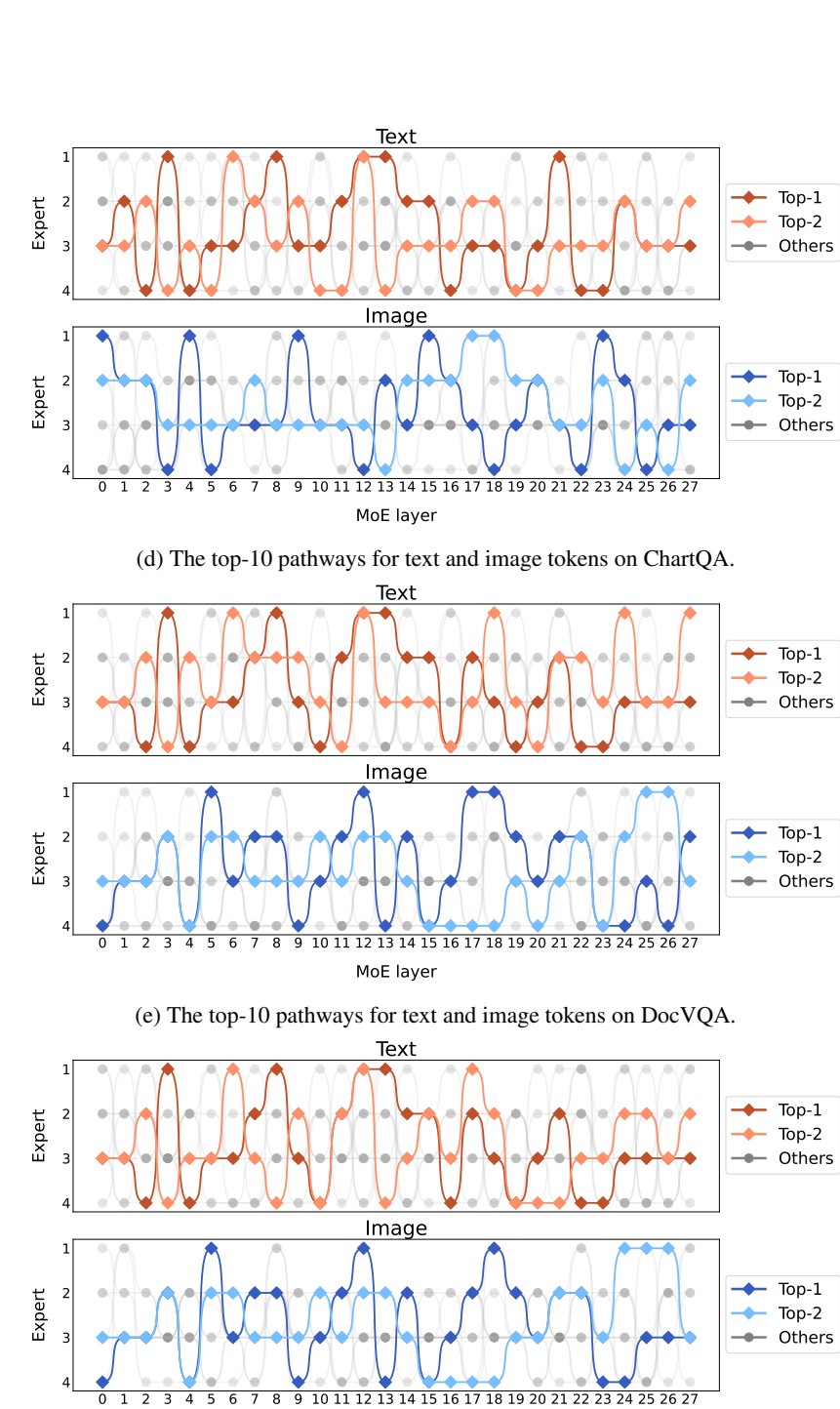

(d) The top-10 pathways for text and image tokens on ChartQA.

(e) The top-10 pathways for text and image tokens on DocVQA.

(f) The top-10 pathways for text and image tokens on InfoVQA.

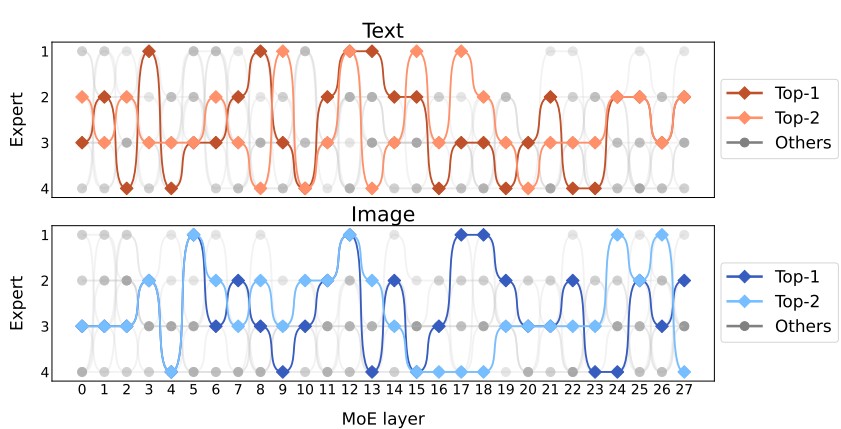

(g) The top-10 pathways for text and image tokens on MMStar.

Figure 5: Visualization of the top-10 activated pathways for text and image modality on various tasks.

