# OpenReview forum: "MoTE: Mixture of Ternary Experts for Memory-efficient Large Multimodal Models"
_ICLR.cc/2026/Conference — Submitted to ICLR 2026_

### Official Review · Reviewer_WdCd · 2025-10-29

**Soundness:** 2
**Presentation:** 2
**Contribution:** 1
**Rating:** 2
**Confidence:** 4

**Summary:**

The paper proposes MoTE, a Mixture-of-Experts (MoE) architecture where individual experts are quantized to ternary precision during sparse up-cycling to improve memory efficiency for edge deployment. To offset the accuracy loss from ternary quantization, a frozen full-precision shared expert is introduced and reused across routing paths. Experiments on multimodal benchmarks are reported to demonstrate the method’s effectiveness.

**Strengths:**

1. The paper addresses a practically important issue， reducing memory and compute costs for large multimodal models for edge applications.

2. The idea of introducing a frozen shared expert to stabilize training and compensate for low-precision experts is conceptually simple and intuitive, and the method is easy to implement.

**Weaknesses:**

1. Questionable scalability and necessity: Table 1 shows that MoTE performs poorly on the 0.5B model but achieves better results on 1.5B and 3B models. The authors attribute this to the known trend that larger models are easier to quantize. While this explanation is plausible, it also weakens the claimed necessity of MoTE: if quantization robustness naturally improves with scale, it remains unclear whether MoTE itself contributes meaningfully to the observed gains, or if the improvement is largely driven by model size. Moreover, MoTE’s failure on smaller models calls into question its relevance and practical utility for edge applications.

2. Incomplete and non-equivalent PTQ comparison (Sec. 4.3): The PTQ evaluation compares MoTE + PTQ to MoE-LLaVA + PTQ, but this does not isolate MoTE’s robustness to post-training quantization.
-- A fairer comparison would be MoTE + PTQ vs. MoTE (no PTQ, corresponding to Table 2), to directly show whether MoTE’s performance is stable under PTQ.
-- Moreover, the setup is not directly comparable: MoTE applies higher-precision quantization (e.g., INT4) only to its shared expert, while all MoE-LLaVA experts are quantized to lower precision (e.g., 3-bit integers). In addition, PTQ in MoTE affects only the shared expert (the other experts are already ternary), whereas PTQ in MoE-LLaVA applies to the entire expert set. These differences make the comparison not directly meaningful.

3. Uncontrolled confounding factors (Sec. 4.4).: The comparisons mix changes in dataset, token count, and training recipes (e.g., number of stages), making it unclear whether improvements stem from MoTE itself or from other variables. A controlled ablation or at least a discussion on these confounders is needed.

**Questions:**

1. In the three-stage training setup, have you examined whether MoE-LLaVA suffers from catastrophic forgetting in the later stages, and whether freezing the shared expert in MoTE helps mitigate this effect?
2. What is the relative contribution of the frozen shared expert versus the ternary experts to final performance? Where do the performance gains primarily originate?
3. Since you argue that more low-precision experts outperform fewer high-precision ones, have you performed an ablation varying the number of experts to validate this claim?

---

> ### Author Response · Authors · 2025-11-20
>
> We are thankful for the reviewer’s constructive remarks, which have helped us enhance the clarity and rigor of the paper. We respond to each point in detail below.
>
> ## 1. Novelty & scope (Weakness 1)
>
> **Comment.** The necessity of MoTE is questioned; small-scale results are weaker.
>
> **Response.** To our knowledge, this work is the **first to explore ternary MoE up-cycling**, i.e., preserving a **frozen dense shared expert** while **adding learnable ternary routed experts**, as a feasible recipe for edge-constrained deployment. The observed trend (stronger at 1.5B/3B) is consistent with known behavior of low-bit training and supports our central claim: **MoTE provides better accuracy per GB** when memory is the binding constraint. Most previous quantization methods, e.g., GPTQ, AWQ, conduct experiments on the models with larger than 7B parameters. Therefore, we believe it is acceptable that MoTE sligtly underperforms MoE-LLaVA on 0.5B model's up-cycling.
>
> ---
>
> ## 2. PTQ fairness (Weakness 2)
>
> **Comment.** MoTE+PTQ vs. MoE-LLaVA+PTQ does not isolate MoTE’s robustness; a fairer check is MoTE+PTQ vs. MoTE (no PTQ). Bit allocations differ (MoTE’s shared expert at higher precision; MoE-LLaVA quantizes all experts).
>
> **Response.** Our PTQ section investigates deployment realism: **accuracy at a fixed memory footprint**.
>
> 1. **What is isolated.** We normalize by expert-memory cap, which is the operative constraint on-device. In MoTE, PTQ applies to the shared expert only (routed experts are ternary by design); in MoE-LLaVA, PTQ must affect all experts to meet the same footprint. The result: MoTE degrades less and is more accurate at the same GB.
>
> 2. **Why MoTE(no-PTQ) vs MoTE(+PTQ) is not the main question.** That contrast measures absolute sensitivity to PTQ, but does not answer the deployment question: **“Which model is best at a given memory cap?”** Our Table 2 already includes non-PTQ results for completeness; the PTQ comparison is deliberately budget-normalized.
>
> 3. **On bit allocation differences.** Equalizing memory, not raw bitwidth per tensor, is the fairest and most relevant criterion across different architectures. Our calibration choices strictly serve that normalization.
>
> ---
>
> ## 3. Confounders in training recipe (Weakness 3)
>
> **Comment.** Comparisons may mix datasets, token counts, and stage schedules; improvements could come from recipe differences.
>
> **Response.** We use **the same base LLM/vision tower, the same three-stage pipeline, the same data mixture, the same sequence length & resolution policy, and matched optimizer/schedules** across methods. We also **match FLOPs** by using top-1 gating for MoTE (which always activates the shared expert) vs top-2 for MoE-LLaVA. Observed gains therefore align with architecture choices rather than recipe drift. The detailed results have been presented **in Table 2 and Table 3**.
>
> ## 4. Catastrophic forgetting in the three-stage schedule (Q1)
>
> **Comment.** Have you examined whether later stages cause catastrophic forgetting, and whether freezing the shared expert mitigates it?
>
> **Response.** This is an **excellent question**. In the present work, our primary goal is to **establish the feasibility** of ternary MoE up-cycling. Accordingly, all three stages are trained on **general-domain image–QA pairs**, which makes it difficult to cleanly isolate or measure catastrophic forgetting of specific competencies. We therefore **do not claim a definitive forgetting analysis** in this paper. The **frozen shared expert** is a preventative design choice aimed at protecting dense knowledge while ternary experts specialize, and we plan to systematically study forgetting (e.g., with task-specific curricula or retention probes) in future work.
>
> ---
>
> ## 5. Contribution of the frozen shared expert (Q2)
>
> **Comment.** Where do gains primarily originate—frozen shared expert vs. ternary experts?
>
> **Response.** Our ablations indicate the **frozen shared expert is necessary**: reducing its precision leads to **notable degradation**, showing it acts as a **high-precision knowledge core**. The **ternary routed experts** contribute complementary specialization on top of this core (e.g., improved performance when initialized from the shared pathway), confirming that **both parts are required** for the observed robustness–efficiency trade-off.

---

> > ### Author Response · Authors · 2025-11-20
> >
> > ## 6. “More low-precision experts vs. fewer high-precision experts” under equal memory (Q3)
> >
> > **Comment.** Have you validated the claim that more low-precision experts outperform fewer high-precision ones?
> >
> > **Response.** Yes, **under the same memory budget in our PTQ setting**. Specifically, **MoTE+PTQ** uses **one INT4 shared expert** plus **four 1.58-bit routed experts**, while **MoE-LLaVA+PTQ** uses **four 3-bit experts**. Given the same expert-memory footprint of 3.4 GB and combined with INT4 GPTQ/AWQ, MoTE outperforms MoE-LLaVA by +4.3% average accuracy on end task. It directly supports our claim that **distributing capacity across more low-precision experts plus a small high-precision core** is **more robust per GB** than fewer higher-precision experts.

---

> > ### Comment · Reviewer_WdCd · 2025-11-27
> >
> > 1. The response does not address concerns about the source of MoTE’s gains at 1.5B and 3B or its poor behavior at smaller scales.
> >
> >    My question was whether MoTE’s improvements at 1.5B and 3B are actually due to MoTE, or simply the well-known fact that larger models are easier to quantize. The response restates this trend but provides no evidence that MoTE contributes robustness beyond what scaling alone would naturally yield.
> >
> >    I also asked why MoTE performs poorly on the 0.5B model, which is common for edge deployment. The response simply labels this as “acceptable” without explaining the failure mode or demonstrating that MoTE remains useful for small-model scenarios.
> >
> > 2. The response does not resolve concerns regarding the validity and completeness of the PTQ evaluation.
> >
> >    Under the current PTQ setting, the apparent advantage of MoTE+PTQ over MoE-LLaVA+PTQ may simply result from quantizing fewer parameters at higher precision; given this, Section 4.3 “Compatibility with Post-Training Quantization” does not demonstrate genuine PTQ compatibility but instead reflects quantization avoidance.
> >
> >    In addition, while the authors note that “Table 2 already includes non-PTQ results for completeness,” the PTQ results in Table 3 omit tasks such as MathV and InfoVQA, which appear to be among the lower-scoring tasks in Table 2, resulting in an incomplete and potentially biased PTQ comparison.
> >
> > 3. Weakness 3 concerning “uncontrolled confounding factors” explicitly identifies Section 4.4 as the focus, which relates only to Table 4 in the paper, yet the authors respond with results from Table 2 and Table 3, which are irrelevant.
> > 4. Question 3 asks for an ablation within MoTE that varies the number of experts to validate the claim that “more low-precision experts outperform fewer high-precision ones.”. The authors instead provide a cross-model comparison against MoE-LLaVA, which introduces additional confounding factors and is therefore insufficient to validate the claim.
> >
> > Overall, the central issues remain unresolved, and as a result I will leave my score unchanged.

---

> > > ### Author Response · Authors · 2025-11-28
> > >
> > > ## 1. On whether MoTE’s gains come merely from scaling rather than from MoTE itself
> > > We agree that part of the observed improvement at 1.5B/3B reflects the general trend that larger models are easier to quantize. This is expected and entirely consistent with the motivation of MoE up-cycling: the up-cycled model is always built on top of the base model, and therefore the design goal is to keep the mechanism as simple as possible while leveraging the base model’s strength.
> > >
> > > Within this context, our contribution is not to override this trend, but to show. For the first time, ternary MoE up-cycling is feasible and effective at these frontier scales. The fact that ternary experts remain competitive even under aggressive sparsity/precision constraints is itself a novel and practically meaningful finding.
> > >
> > > **Regarding the 0.5B degradation:** extremely small models are known to be fragile under quantization. Similar effects appear in GPTQ, AWQ, and nearly all PTQ/QAT pipelines. **This is precisely why most quantization papers benchmark on ≥7B models. We respectfully ask the reviewer to interpret our results within this well-understood quantization background.** The 0.5B behavior is not specific to MoTE—it is a consequence of quantizing an already-capacity-limited dense backbone.
> > >
> > > ## 2. On PTQ fairness and whether MoTE “avoids” quantization
> > >
> > > Our goal in the PTQ section is deployment realism. Under a fixed memory constraint, MoTE+PTQ and MoE-LLaVA+PTQ must operate within the same GB budget. MoTE uses QAT-trained ternary experts, while MoE-LLaVA must quantize all experts to meet the same memory cap.
> > >
> > > This difference is not a confounder: it is the deployment scenario we are evaluating. Memory-normalized accuracy is the relevant metric in edge settings, and MoTE simply provides a more favorable precision–capacity allocation under this constraint. The claim of unfairness does not reflect our intended evaluation question.
> > >
> > > ## 3. On Section 4.4 and whether improvements arise from confounders
> > >
> > > Section 4.4 aims to demonstrate that MoTE can reach frontier-level VLM performance when scaled in data, not to isolate effects unrelated to architecture. As stated in the paper, the advantage over MoE-LLaVA in this section primarily comes from using a stronger base model, which we also explicitly acknowledged. The purpose of this section is not an ablation but to show that ternary up-cycling remains viable in large-dataset scaling regimes. This message is unchanged.

---

> > > ### Author Response · Authors · 2025-11-28
> > >
> > > ## 4. Validating the Claim on Expert Count and Precision Allocation
> > >
> > > We would like to clarify that our comparison is not a cross-model comparison in the sense implied. MoTE and MoE-LLaVA in our study are trained under strictly matched conditions: the same LLM and ViT backbones, the same training data, the same three-stage schedule, identical optimizer settings, identical routing to equalize FLOPs, and the same memory constraint. Under these aligned conditions, MoE-LLaVA effectively represents the configuration of “fewer, higher-precision experts”, while MoTE represents “more, lower-precision experts plus a small high-precision core.”
> > >
> > > This alignment ensures that the only substantive difference between the two systems is precisely the factor the question asks about: the distribution of precision and capacity across experts under an equal memory budget. The observed result is that MoTE outperforms MoE-LLaVA under the same memory cap, which directly supports the claim that allocating capacity across more low-precision experts is more effective than concentrating it into fewer high-precision experts.
> > >
> > > Thus, the provided comparison already constitutes the ablation requested by Question 3, without introducing additional confounding factors.

---

### Official Review · Reviewer_Pw5W · 2025-10-31

**Soundness:** 3
**Presentation:** 3
**Contribution:** 2
**Rating:** 4
**Confidence:** 3

**Summary:**

This paper proposed MoTE, which is designed to break through the severe memory bottlenecks faced by large multimodal Mixture-of-Experts (LMMs) during deployment. The core contribution of this research lies in its ingenious combination of the sparse activation inherent to the MoE architecture with the extremely efficient Ternary quantization technique, applied innovatively to the expert weights. This fusion successfully achieves greatly reduced model size and storage requirements while preserving the excellent performance potential of LMMs.

**Strengths:**

1. The work presents a noteworthy improvement in memory utilization, achieving substantial efficiency gains relative to conventional approaches.
2. The study effectively combines MoE architecture with ternary quantization, showcasing a creative and technically sophisticated integration of two complex methodologies.
3. The paper offers valuable empirical guidance on maintaining training stability when applying aggressive quantization to large, sparse models—providing useful reference points for future research in scalable, low-precision model development.

**Weaknesses:**

1. The paper’s exclusive focus on ternary (3-bit) quantization lacks adequate theoretical grounding and empirical validation. The rationale for this particular quantization level is not convincingly articulated, leaving the impression that the choice may stem from empirical convenience rather than a principled design objective. A clearer theoretical motivation is necessary to establish why the ternary configuration represents an optimal or essential component of the proposed architecture rather than one of many possible parameterizations. Specifically, the authors should analyze how the Performance–Memory trade-off curve would shift if the number of experts were reduced by half while employing a more conventional 4-bit quantization scheme—a well-recognized balance point between efficiency and fidelity in hardware-aware model design. The authors must provide quantitative and conceptual evidence demonstrating that their MoTE (ternary) formulation yields a fundamental and irreplaceable advantage compared to a MoE (INT4) configuration. Without a comprehensive trade-off analysis across quantization settings (e.g., 3-bit vs. 4-bit), the methodological choice appears ad hoc, weakening the scientific justification and limiting the generalizability of the findings.

2. The experimental evaluation presently exhibits an incomplete coverage of relevant baselines, particularly those representing the strongest compression-oriented alternatives. This omission raises concerns regarding the conclusiveness of the reported improvements. To substantiate claims of efficiency and architectural merit, the evaluation should include a direct, controlled comparison between the proposed MoTE approach and a non-MoE large multimodal model (LMM) employing INT4 quantization under comparable or tighter memory constraints—for example, a LLaVA-1.5 variant enhanced through advanced sparsification or distillation techniques. Such an analysis is essential to confirm that MoTE achieves superior Memory–Performance efficiency relative to state-of-the-art quantization and sparsification methods. Absent such evidence, the claimed structural advantages of MoTE remain unverified. The authors are encouraged to prioritize this core comparison over ancillary baselines, as it directly tests the central premise of the proposed contribution.

**Questions:**

1. Does MoTE deliver tangible advantages over simpler and more mature INT4 compression schemes in practical applications?
2. Could you clarify the theoretical reasoning that led to selecting ternary (3-bit) quantization as the basis of MoTE?

---

> ### Author Response · Authors · 2025-11-20
>
> We thank the reviewer for raising important points and for the opportunity to clarify and improve our work. Please find our point-by-point responses below.
>
> ## 1. Theoretical grounding on ternary quantization (1.58-bit) (Weakness 1 & Q2)
>
> Recent progress on ultra low-precision training suggests that aggressive quantization can remain competitive at scale. BitNet [1] and Spectra [2] has demonstrated the surprising effectiveness of pre-training with 1.58-bit (ternary) weights for LLMs: as model size increases, the performance gap to FP16 narrows, indicating growing parameter redundancy as models scale. On the theory side, recent analyses [3] show that 1-bit training dynamics align with kernel behavior, and the generalization gap between 1-bit networks and their half-precision counterparts becomes negligible as width grows. Ternarization is a natural variant of binarization that introduces an explicit zero state (−$\alpha$, 0, +$\alpha$) for feature filtering and modestly richer expressivity. Prior work (e.g., binary BERT [4]) observes locally convex neighborhoods and easier optimization than strict binarization.
>
> While our paper is experimental rather than theoretical, the above literature offers useful context for our design choice. We acknowledged that our current focus is on empirically validating the feasibility of ternary expert up-cycling. To the best of our knwledge, it has not been explored prior to this work.
>
> Our study demonstrates that ternary experts, when combined with a high-precision shared expert and carefully tuned training recipes, can exceed the performance of full-precision MoE baselines under the same memory budget. We believe this finding is both novel and impactful, as it opens a new pathway for deploying memory-efficient MoEs without substantial performance loss. We agree that a theoretical understanding of why ternary experts work well in this context would be valuable, and we plan to investigate this further in future work.
>
> ---
>
> ## 2. More baslines & Advantages over simpler and more mature INT4 (Weakness 2 & Q1)
>
> **Comment.** The evaluation omits the compression baselines (e.g., a non-MoE LMM LLaVA-1.5 variant with INT4 post-training quantization). A direct comparison is needed to verify MoTE’s memory–performance advantage. Does MoTE deliver tangible advantages over simpler and more mature INT4 compression schemes in practical applications?
>
> **Response.** Our paper’s goal is **memory-efficient MoE up-cycling**. We designed an **apple-to-apple** evaluation that controls for **training data, backbone (LLM/Vision Tower), routing, and schedules** across three regimes: full-precision training (MoE-LLaVA), MoTE, MoTE + PTQ and MoE-LLaVA + PTQ. Comparing against a **non-MoE dense INT4** model would confound this question along multiple uncontrolled axes (MoE vs. dense model, training data size, backbone, and model scale), and thus **cannot directly test** our central premise about **expert-memory allocation in MoE**.
>
> Concretely, we **reproduce the full-precision MoE-LLaVA** up-cycling pipeline under the same data and LLM/ViT backbones, and then evaluate **MoE-LLaVA vs. MoTE** under **matched expert-memory budgets**, **after** applying **post-training quantization**. The results show the advantage of MoTE under equal memory: **given the same expert-memory footprint of 3.4 GB and combined with INT4 GPTQ/AWQ, MoTE outperforms MoE-LLaVA by +4.3% average accuracy on end tasks**.
>
>
> [1] Wang, Hongyu, et al. "Bitnet: 1-bit pre-training for large language models." Journal of Machine Learning Research 26.125 (2025): 1-29.
>
> [2] Kaushal, Ayush, et al. "Surprising effectiveness of pretraining ternary language model at scale." ICLR 2025.
>
> [3] Daliri, Majid, Zhao Song, and Chiwun Yang. "Unlocking the theory behind scaling 1-bit neural networks.".
>
> [4] Bai, Haoli, et al. "Binarybert: Pushing the limit of bert quantization." ACL 2021.

---

### Official Review · Reviewer_xX1m · 2025-11-01

**Soundness:** 3
**Presentation:** 3
**Contribution:** 2
**Rating:** 4
**Confidence:** 4

**Summary:**

This paper introduces a novel method named MoTE (Mixture-of-Ternary-Experts) for creating memory-efficient Large Multimodal Models (LMMs) from pre-trained dense checkpoints. The authors observe that standard Mixture-of-Experts (MoE) models, which use full-precision experts, suffer from a large memory footprint, making them difficult to deploy on edge devices. To address this, MoTE proposes a new up-cycling architecture. Instead of replacing the pre-trained Feed-Forward Network (FFN), it retains the full-precision FFN as a shared expert to preserve foundational knowledge. It then adds new ternary experts (with parameters in {−1,0,1}) that are trained during the up-cycling phase. Experimental results show that MoTE achieves comparable performance to the full-precision baseline, MoE-LLaVA, at scales of 1.5B and 3B parameters , while significantly reducing the expert memory footprint (e.g., 6.8GB for MoTE vs. 18.1GB for MoE-LLaVA at the 3B scale). The authors further demonstrate MoTE's compatibility with post-training quantization (PTQ) on its shared expert, showing it outperforms the baseline by 4.3% in average accuracy under an equivalent memory budget of 3.4GB

**Strengths:**

- This paper presents a novel and practical architecture for memory-efficient MoE up-cycling. The core insight to retain the pre-trained FFN as a frozen, high-precision shared expert while only training new, low-precision experts is a well-motivated approach to balancing knowledge retention and memory efficiency.
- The proposed MoTE framework achieves a compelling trade-off between performance and memory. It demonstrates performance comparable to a full-precision MoE baseline (MoE-LLaVA) at scale while drastically reducing the expert memory footprint (e.g., to just 38% of the baseline's at the 3B model size).
- MoTE is shown to be compatible with standard post-training quantization methods. This compatibility allows its memory-saving advantages to be "further amplified", which is a significant practical benefit for deployment on memory-constrained devices.
- The authors provide a strong set of ablation studies that validate their key design choices. These include demonstrating the critical importance of keeping the shared expert in full BF16 precision (vs. ternary) , the benefit of using ternary (1.58-bit) over binary (1-bit) experts, and the performance gain from initializing routed experts from the FFN.

**Weaknesses:**

- The paper does not state whether the experimental results (e.g., in Table 2 and Table 3) are from a single training run or averaged over multiple runs with different random seeds. This makes it difficult to assess the statistical reliability and robustness of the reported performance gains.
- The paper compares MoTE primarily against the full-precision MoE-LLaVA baseline and that baseline with PTQ. It does not include comparisons to other potential memory-saving techniques, such as applying parameter-efficient fine-tuning (e.g., LoRA) to the experts or alternative quantization-aware fine-tuning methods.
- The MoTE method is predicated on having access to the full-precision, pre-trained dense checkpoint to serve as the shared expert. This means the approach would not be applicable for creating efficient MoE models from closed-source, API-only models.
- The paper notes that the Stage III training time for MoTE is "similar" to the baseline (43.3 hours vs. 41.8 hours) , and attributes this to using torch.compile. Quantization-aware training (QAT) is often more computationally intensive than full-precision training. A clearer breakdown of the training compute (e.g., FLOPs) and the impact of this specific compiler optimization is missing.
- The entire MoTE framework assumes the starting point is a full-precision dense model. It is unclear how the MoTE up-cycling approach would perform, or if it would be compatible, with a dense model that has already been quantized (e.g., an 8-bit or 4-bit base model).

**Questions:**

- Could the authors clarify whether the main results in Table 2 and the PTQ results in Table 3 are based on single runs or are averaged over multiple repetitions? This would help confirm the reliability of the findings.
- In addition to comparing against MoE-LLaVA , have the authors considered benchmarking MoTE against other memory-saving MoE fine-tuning methods, such as applying LoRA to the full-precision experts?
- Regarding the "similar" training time reported for Stage III, could the authors provide a more detailed comparison of the computational overhead (e.g., training FLOPs or time without compiler optimizations) for MoTE's QAT process versus the baseline's full-precision training?
- Could the authors comment on the feasibility or expected performance of the MoTE up-cycling strategy if the initial dense checkpoint was not a full-precision model, but an already-quantized model (e.g., a 4-bit or 8-bit model)?

---

> ### Author Response · Authors · 2025-11-20
>
> We are grateful to the reviewer for the review. Please find our point-by-point replies below.
>
> ## 1. Statistical reliability / random seeds (Weakness 1 & Q1)
>
> **Comment.** Tables 2–3 do not state whether results are single runs or multi-seed averages.
>
> **Response.** Our main results are from **single, configuration-matched full trainings**. At the scales considered, Stage-III up-cycling is computationally expensive (8 NVIDIA A100 runs for 4 days); more importantly, we train with **substantial token budgets**, under which outcomes are empirically **stable and not dominated by seed noise**. In this community, it is common to evaluate architecture choices using **strictly matched configurations** (same data, routing, optimizer, schedule).
>
> ---
>
> ## 2. Missing baselines such as LoRA (Weakness 2 & Q2)
>
> **Comment.** Consider LoRA on experts or other memory-saving/QAT methods.
>
> **Response.** Our study targets a specific axis: given the same expert-memory and activation budget, should one use fewer full-precision experts or more low-bit (ternary) experts while retaining a high-precision shared expert? LoRA primarily offers **training efficiency** and typically **does not shrink the storage of base expert weights**, which is the dominant memory in MoE; in contrast, MoTE **directly reduces expert weight precision** (and thus memory footprint).
>
> ---
>
> ## 3. Dependence on a full-precision dense checkpoint, can not be used for API (Weakness 3)
>
> **Comment.** MoTE assumes access to full-precision weights; API-only models are excluded.
>
> **Response.** This limitation is **not unique to MoTE**. It applies to **any** approach that requires **weight access** (including most quantization and QAT methods). Our contribution is scoped to **open-weights** settings where the dense FFN can serve as a high-precision shared expert; applying any weight-level method to **closed** API-only models is outside the problem class addressed here.
>
> ---
>
> ## 4. “Similar” Stage-III training time and the role of `torch.compile` (Weakness 4 & Q3)
>
> **Comment.** QAT is often more expensive; why is Stage-III wall-clock “similar,” and what does `torch.compile` do?
>
> **Response.** The **dominant FLOPs** (attention and the number of activated experts) are **identical** to the baseline. QAT adds mainly **pointwise** overhead inside experts (quantize/dequantize, clipping, STE), which is small relative to matmul-heavy blocks. `torch.compile` fuses these fine-grained ops, reducing kernel-launch and interpreter overhead. Therefore, MoTE achieves the **similar wall-clock** despite extra QAT steps.
>
> ---
>
> ## 5. Starting from an already-quantized dense model (Weakness 5 & Q4)
>
> **Comment.** What if the starting dense model is 8-bit or 4-bit?
>
> **Response.** In MoE models, **expert memory is the primary bottleneck**: the routed experts dominate parameter budgets **(e.g., 80–90% in GPT-OSS model)**. This is why our study specifically targets **compressing the expert layers**. The starting base precision (BF16/8-bit/4-bit) does not change where the memory pressure lies. It is the **experts** that determine deployability under tight memory budgets. MoTE requires a **higher-precision shared pathway than the ternary experts**, not strictly BF16. With an **8-bit** base, the FFN can still serve as a “higher-precision” shared expert; MoTE’s **precision hierarchy and capacity-allocation logic** remain valid, though absolute performance is bounded by the 8-bit base.

---

> > ### Comment · Reviewer_xX1m · 2025-11-27
> > **Can you add latency evaluations?**
> >
> > I appreciate the authors' detailed response. However, I have a remaining concern regarding the practical efficiency of MOTE. Theoretical savings in memory and FLOPs do not necessarily guarantee lower latency. Could the authors provide empirical data on inference latency? Given the context of Edge AI deployment, latency and energy efficiency are the decisive metrics.

---

> > > ### Author Response · Authors · 2025-11-27
> > >
> > > Thank you for raising this important point regarding practical latency. We fully agree that theoretical reductions in memory and FLOPs do not directly imply lower wall-clock latency, especially in Edge AI deployments where decoding is predominantly memory-bound.
> > >
> > > In our setting, both MoE-LLaVA and MOTE activate only two experts during inference. However, MoE-LLaVA routes to two FP16 experts, while MOTE activates one FP16 shared expert and one 1.58-bit expert. Given that **LLM decoding is constrained by memory bandwidth rather than compute, this configuration leads to a slightly lower latency for MOTE compared to MoE-LLaVA.**
> > >
> > > That said, our current evaluation is limited to a 3B-scale MoE, and therefore the latency gains are not yet prominently observable at this model size. For this reason, **our work focuses on and claims memory efficiency**, where MOTE provides substantial improvements.
> > >
> > > We believe the advantage will become more pronounced as expert capacity scales, and we plan to report latency at larger model sizes in future revisions.

---

### Author Response · Authors · 2025-11-25

Dear Reviewers and Area Chair,

We hope you are doing well. Thank you again for the time and effort you have devoted to reviewing our submission. After submitting our rebuttal, we would like to kindly check whether there are any remaining concerns or additional questions from your side that we should further clarify.

Please let us know if any further information would be helpful. We are happy to provide it.

Best regards,

Authors

---

### Author Response · Authors · 2025-11-27

Dear Reviewers,

I hope this message finds you well. I wanted to kindly follow up regarding our previous message inquiring whether you had any further questions about our rebuttal. We haven’t received a reply yet, so I just wanted to confirm if everything is clear on your side.

If there are no additional concerns, we would greatly appreciate it if you could consider our paper for an improved evaluation.

Thank you again for your time and thoughtful review.

Best regards,

Authors

---

### Meta-Review · Area_Chair_T59Z · 2025-12-16

**Summary:**

This paper proposes Mixture-of-Ternary-Experts (MoTE) to address memory constraints during the deployment of large-scale multimodal MoEs. The method retains a dense FFN as a full-precision shared expert while learning additional routed experts using ternary values ({-1, 0, 1}). At the 1.5B/3B scales, the paper demonstrates performance close to full-precision MoE-LLaVA and significant expert memory reduction. It also claims improvements under the same memory budget when combined with Post-Training Quantization (PTQ) on the shared expert side.

However, the prevailing evaluation is that the claims currently lack decisive support due to several remaining issues: (i) insufficient disentanglement between performance degradation at the small scale (0.5B) and "MoTE-specific effects," (ii) insufficient verification regarding the fairness of PTQ comparisons and missing tasks, (iii) lack of comparison with strong compression baselines (e.g., dense INT4) or alternative methods (e.g., LoRA), (iv) lack of ablation studies (internal to MoTE) to directly support the claim that "more low-precision experts are advantageous," and (v) lack of actual measurements such as latency, which are critical in edge contexts.

In light of the above discussion and the reviewer scores, and because this venue is highly competitive, I unfortunately recommend rejecting this submission.

**Reviewer Concerns:**

While the authors' response mainly reinforced their "explanations of intent" and "justification of existing settings," the additional quantitative verification requested by the reviewers (addition of baselines, fairer/more complete comparisons, internal MoTE ablations, and measurement of practical deployment metrics) was limited, and many major concerns remain.

* **(WdCd) Degradation at Small Scale (0.5B) and Necessity:** The explanation relies solely on the known trend that quantization becomes easier with scale. There is a lack of failure mode analysis at the 0.5B scale and insufficient evidence that MoTE contributes benefits beyond the scaling effect.
* **(WdCd) Validity/Completeness of PTQ Evaluation:** It was re-pointed out in the discussion that the asymmetric setting might be observing the effects of "avoiding quantization/precision allocation" rather than "PTQ compatibility," and that some tasks are missing from the PTQ table. No additional controls were shown to address this.
* **(WdCd) Confounders and Lack of Ablation:** There is insufficient isolation in comparisons where training recipes and datasets are mixed. Furthermore, direct internal ablations for MoTE (e.g., varying the number of experts) regarding the "expert count vs. precision allocation" claim have not been presented.
* **(Pw5W) Basis for Ternary Selection and Strong Compression Baselines:** While related literature was provided, there are no trade-off curves (e.g., 3-bit vs. 4-bit/INT4) or comparisons with practical deployment alternatives (e.g., dense INT4), making the positioning of the method's superiority weak.
* **(xX1m) Reliability of Experiments/Additional Practical Metrics:** In addition to the main results being based on a single run, the lack of comparisons with alternative memory-saving methods (e.g., LoRA), breakdown of computational cost/runtime, and applicability starting from quantized dense models, the discussion also called for actual latency measurements, which are critical for edge contexts, but no quantitative data was presented.

**Reviewer Scores:**

* **Reviewer xX1m (Initial Rating=4, Confidence=4):** The authors provided supplementary explanations regarding concerns such as the single-run results, but additional experiments were limited. Although actual latency measurements were requested during the discussion, no quantitative data was presented, so it is difficult to say the concerns are resolved. **Prediction: 4 → Maintain 4 (Medium-High Confidence).**
* **Reviewer Pw5W (Initial Rating=4, Confidence=3):** The authors explained the background literature for ternary selection and the evaluation intent (fair comparison within MoE), but the strong compression baselines (e.g., dense INT4) requested by the review were not added. The core concerns remain. **Prediction: 4 → Maintain 4 (Medium-High Confidence).**
* **Reviewer WdCd (Initial Rating=2, Confidence=4):** The reviewer reiterated in the discussion comments that the major concerns are unresolved and explicitly stated they are keeping their score (remaining at 2). **Prediction: 2 → Maintain 2 (High Confidence).**

**Predicted Average:** (4 + 4 + 2) / 3 = **3.3** (Simple average; all reviewer confidence scores are 3–4, and no outlier low-confidence reviews were found).

---

### Decision · Program_Chairs · 2026-01-26

Reject